Spider crabs of the Western Atlantic with special reference to fossil and some modern Mithracidae

Klompmaker Adiël A. 1 2 adielklompmaker@gmail.com
Portell Roger W. 1
Klier Aaron T. 3
Prueter Vanessa 3
Tucker Alyssa L. 2
1 Florida Museum of Natural History, University of Florida , Gainesville, FL , United States of America
2 Department of Geological Sciences, University of Florida , Gainesville, FL , United States of America
3 Department of Biology, University of Florida , Gainesville, FL , United States of America
De Baets Kenneth
Electronic publication date: 2015 Oct 1
Publication date: 2015
Volume: 3
Electronic Location ID: e1301
Received 2015 Aug 5; Accepted 2015 Sep 16
Copyright: © 2015 Klompmaker et al.
Copyright year: 2015
Copyright holder: Klompmaker et al.
License: This is an open access article distributed under the terms of the Creative Commons Attribution License, which permits unrestricted use, distribution, reproduction and adaptation in any medium and for any purpose provided that it is properly attributed. For attribution, the original author(s), title, publication source (PeerJ) and either DOI or URL of the article must be cited.
License URL: https://creativecommons.org/licenses/by/4.0/

Keywords: Biodiversity, Paleontology, Decapoda, Crustacea, Biogeography, Taxonomy, Majoidea, Systematics, Paleoecology, Reef

Funding: Jon L. and Beverly A. Thompson Endowment Fund This work was supported by the Jon L. and Beverly A. Thompson Endowment Fund to AAK. The funders had no role in study design, data collection and analysis, decision to publish, or preparation of the manuscript.

==============================
Spider crabs (Majoidea) are well-known from modern oceans and are also common in the western part of the Atlantic Ocean. When spider crabs appeared in the Western Atlantic in deep time, and when they became diverse, hinges on their fossil record. By reviewing their fossil record, we show that (1) spider crabs first appeared in the Western Atlantic in the Late Cretaceous, (2) they became common since the Miocene, and (3) most species and genera are found in the Caribbean region from the Miocene onwards. Furthermore, taxonomic work on some modern and fossil Mithracidae, a family that might have originated in the Western Atlantic, was conducted. Specifically, Maguimithrax gen. nov. is erected to accommodate the extant species Damithrax spinosissimus, while Damithrax cf. pleuracanthus is recognized for the first time from the fossil record (late Pliocene–early Pleistocene, Florida, USA). Furthermore, two new species are described from the lower Miocene coral-associated limestones of Jamaica (Mithrax arawakum sp. nov. and Nemausa windsorae sp. nov.). Spurred by a recent revision of the subfamily, two known species from the same deposits are refigured and transferred to new genera: Mithrax donovani to Nemausa, and Mithrax unguis to Damithrax. The diverse assemblage of decapods from these coral-associated limestones underlines the importance of reefs for the abundance and diversity of decapods in deep time. Finally, we quantitatively show that these crabs possess allometric growth in that length/width ratios drop as specimens grow, a factor that is not always taken into account while describing and comparing among taxa.

Introduction

Modern spider crabs (Majoidea) range in size from a few millimeters to more than a meter in carapace length (Griffin, 1966). Long, slender legs and a pyriform to triangular shape give many of them a spider-like appearance. They occur in nearly all oceans (e.g., Griffin, 1966), and many of them have been found to decorate themselves for camouflage (e.g., Wicksten, 1993; Guinot, Tavares & Castro, 2013). Today, spider crabs are very diverse with nearly 1,000 species worldwide (Ng, Guinot & Davie, 2008; De Grave et al., 2009). More than 125 species have been found in the fossil record (De Grave et al., 2009; Schweitzer et al., 2010), with the oldest species known from the mid-Cretaceous of Europe (Breton, 2009; Klompmaker, 2013). Collins, Portell & Donovan (2009) provided an overview of fossil decapods, including majoids, known from the Caribbean region. Since then, various new fossil majoid occurrences and new fossil species have been reported for the Western Atlantic (e.g., Collins et al., 2010; Collins & Donovan, 2012; Feldmann et al., 2013; Franţescu, 2013; Varela, 2013; Collins, Garvie & Mellish, 2014; Stepp, 2014).

The Mithracidae sensu Windsor, 2010, and Windsor & Felder, 2014 (=Mithracinae sensu Ng, Guinot & Davie, 2008, and De Grave et al., 2009) are found in (sub)tropical waters from intertidal to 450 m depth, mainly as reef- and rubble dwellers (Windsor & Felder, 2014). Recently, the family was revised extensively using morphological and molecular analyses resulting in numerous redefinitions and the resurrection and erection of four genera (Windsor & Felder, 2014). As is the case for the Western and Eastern Pacific, the Mithracidae are well-known from the Western Atlantic with over 30 species (e.g., Rathbun, 1925; Abele & Kim, 1986; Felder et al., 2009; Alves et al., 2012; Windsor & Felder, 2014), the latter authors (p. 154) suggesting it is an “amphi-American” group. Although their fossil record is decent with 25 known species (Schweitzer et al., 2010; Tables 1 and S1), mostly originating from the Western Atlantic (Table 1), additional research is required because representatives of many extant genera have a scarce fossil record.

Table 1 The number of reported fossil carapace specimens of species of Mithracidae worldwide and their country of origin and age.

Note that Windsor & Felder (2014) ascribed Stenocionops Desmarest, 1823, and Micippa Leach, 1817, to Mithracidae s.l.; the fossil Antarctomithrax Feldmann, 1994, was not discussed.

Taxon	Number of fossil carapace specimens	Country of origin and age	References other than original description	
*Antarctomithrax thomsoni Feldmann, 1994	1	Antarctica [Eocene]		
Damithrax cf. pleuracanthus (Stimpson, 1871)	1	USA (Florida) [late Pliocene—early Pleistocene]	herein	
*Damithrax unguis (Portell & Collins, 2004)	24	Jamaica [early Miocene]	herein	
Maguimithrax spinosissimus (Lamarck, 1818)	1	Barbados [Pliocene, Pleistocene], Curaçao [Pliocene], Jamaica [late Pleistocene]	Collins & Morris (1976); Morris (1993); Stepp (2014)	
*Micippa annamariae Gatt & De Angeli, 2010	3	Malta [late Miocene]		
*Micippa antiqua Beschin, De Angeli & Checchi, 2001	2	Italy [early Oligocene]	De Angeli & Beschin (2008)	
*Micippa hungarica (Lörenthey in Lörenthey & Beurlen, 1929) [=Maia austriaca Bachmayer, 1953; Phrynolambrus weinfurteri Bachmayer, 1953]	70	Austria [late Miocene], Hungary [middle Miocene], Poland [middle Miocene]	Bachmayer (1953); Müller (1974); Müller (1984); Müller (1996)	
Micippa cf. thalia (Herbst, 1782–1804)	0 (rostral spines)	Japan [Pleistocene]	Kato & Karasawa (1998)	
Mithraculus aff. coryphe (Herbst, 1782–1804)	1	Jamaica [late Miocene]	Collins et al. (2010)	
Mithraculus cf. forceps Milne-Edwards, 1873–1880	5	Jamaica [late Pleistocene]	Morris (1993); Collins, Donovan & Stemann (2009)	
Mithrax aculeatus (Herbst, 1782–1804) (=Mithrax verrucosus Milne-Edwards, 1832)	5	Barbados [Pliocene, Pleistocene], Jamaica [late Pleistocene]	Collins & Morris (1976); Morris (1993); Collins, Donovan & Stemann (2009)	
*Mithrax arawakum sp. nov.	2	Jamaica [early Miocene]	herein	
Mithrax hemphilli Rathbun, 1892	6	Barbados [Pleistocene]	Collins & Morris (1976)	
Mithrax “hispidus” (Herbst, 1782–1804) [=M. caribbaeus Rathbun, 1920]	7	Barbados [Pliocene, Pleistocene], Cuba [Plio-Pleistocene], Jamaica [late Pleistocene]	Collins & Morris (1976); Morris (1993); Collins, Donovan & Dixon (1996); Varela & Rojas-Consuegra (2009); Varela & Rojas-Consuegra (2011)	
Nemausa acuticornis (Stimpson, 1871)	2	Jamaica [late Pleistocene]	Collins, Donovan & Stemann (2009)	
*Nemausa donovani (Portell & Collins, 2004)	1	Jamaica [early Miocene]	herein	
*Nemausa windsorae sp. nov.	1	Jamaica [early Miocene]	herein	
Stenocionops coelatus (Milne-Edwards, 1878)	0 (dactylus and propodus)	USA (North Carolina) [early Pliocene]	Young (2014)	
*Stenocionops dyeri Blow, 2003	7	USA (Virginia) [Pliocene]	herein	
*Stenocionops primus Rathbun, 1935	0 (propodus)	USA (Arkansas) [Santonian]		
*Stenocionops suwanneeana Rathbun, 1935	0 (propodus)	USA (Florida) [late Eocene]	Portell (2004)	
*Teleophrys acornis Portell & Collins, 2004	1	Jamaica [early Miocene]		
Teleophrys ruber (Stimpson, 1871)	1	Barbados [Pliocene]	Collins & Morris (1976)	
*Thoe asperoides Collins & Todd (in Todd & Collins, 2005)	1	Panama [late Miocene], Costa Rica [late Pliocene—early Pleistocene]		
*Thoe vanuaensis (Rathbun, 1945)	0 (left chela)	Fiji [Pliocene]		
Notes.

* Denotes exclusively fossil mithracid species.

Their limited fossil record is expressed clearly in the low number of specimens available for fossil Mithracidae. Table 1 shows that, on average, only about six carapaces are available per species for those that have a fossil record, with the average being highly skewed by Damithrax unguis (Portell & Collins, 2004) and Micippa hungarica (Lörenthey in Lörenthey & Beurlen, 1929). In fact, 18/25 species are represented by no more than three carapaces. This severely hampers our understanding of intraspecific variation, possible sexual dimorphism, and ontogenetic variation, all of which are important factors when defining a species. As quantitatively shown, carapace length/width ratios are known to change throughout ontogeny for a variety of fossil crabs (e.g., Klompmaker, Feldmann & Schweitzer, 2012; De Jesús Gómez-Cruz, Bermúdez & Vega, 2015), and allometric growth was recently found for fossil ghost shrimp claws as well (e.g., Klompmaker et al., 2015). Although recent diagnoses of fossil crab genera and species usually contain statements about length/width ratios (e.g., Fraaije et al., 2010; Feldmann et al., 2011; Hyžný & Schlögl, 2011; Klompmaker, Artal & Gulisano, 2011; Klompmaker et al., 2011; Schweitzer & Feldmann, 2011; Luque et al., 2012; Starzyk, Krzemińska & Krzemiński, 2012; Vega et al., 2012; Artal, Van Bakel & Onetti, 2014; Ossó & Díaz Isa, 2014; Kočová Veselská, Kočí & Kubajko, 2014; Beschin, Busulini & Tessier, 2015), possible changes to this ratio as the animal grows were not always studied, in part due to a limited number of specimens. Possible allometric growth is also important to evaluate for genera of Mithracidae that are currently diagnosed, in part, based on carapace length/width ratios (Windsor & Felder, 2014). The same applies for mithracid species as, for example, allometric growth associated with reproduction was recorded for a modern mithracid species (Cobo & Alves, 2009). Here, we review the fossil record of spider crabs (Majoidea) in the Western Atlantic to elucidate their occurrences through time and their paleobiogeography. Furthermore, various fossil and modern members of the Mithracidae are described or reassigned in the Systematic Paleontology section, and allometric growth of these majoids is discussed because a sufficient number of specimens are available for several species to do so.

Materials & Methods

We compiled data on all fossil majoid occurrences known from the Western Atlantic (defined here: Argentina to Canada) determined to the genus- and species-levels based on the literature and previously unreported material from the FLMNH Invertebrate Paleontology Collection. Possible majoids not determined to at least the genus-level (e.g., Vega, Feldmann & Sour-Tovar, 1995; Kornecki, 2014; FLMNH IP Collection) were not included.

For the systematics part, the length and width of crab carapaces were measured with digital calipers accurate to 0.03 mm and the ratio between length and width was calculated where possible. Institutional abbreviations for specimens: FSBC: Fish and Wildlife Research Institute, St. Petersburg, Florida, USA; UF: Florida Museum of Natural History at the University of Florida, Gainesville, Florida, USA. Modern UF specimens are housed in Invertebrate Zoology (IZ); fossil specimens in Invertebrate Paleontology (IP).

The electronic version of this article in Portable Document Format (PDF) will represent a published work according to the International Commission on Zoological Nomenclature (ICZN), and hence the new names contained in the electronic version are effectively published under that Code from the electronic edition alone. This published work and the nomenclatural acts it contains have been registered in ZooBank, the online registration system for the ICZN. The ZooBank LSIDs (Life Science Identifiers) can be resolved and the associated information viewed through any standard web browser by appending the LSID to the prefix http://zoobank.org/. The LSID for this publication is: urn:lsid:zoobank.org:pub:6049E531-ABA7-43EA-8308-EEB5029F667F. The online version of this work is archived and available from the following digital repositories: PeerJ, PubMed Central and CLOCKSS.

Results

Spider crab distribution in the Western Atlantic

Genera and species known today (15/19 genera or 79%, 14/32 species or 44%) are well-represented in the dataset (Table S1) on fossil spider crabs because most taxon occurrences (>85%) are Neogene and Quaternary in age. Spider crabs in this part of the world first appeared the late Late Cretaceous (Rathbun, 1935; Feldmann et al., 2013), which is younger than the mid-Cretaceous occurrences in Europe (Breton, 2009; Klompmaker, 2013). They become increasingly better represented towards the Recent on the genus- and family-levels (Fig. 1, Tables 1 and S1). For example, two majoid genera are reported from the Late Cretaceous, none from the Paleocene, four from the Eocene, two from the late Oligocene—early Miocene, nine from the Miocene, ten from the Pliocene, and eight from the Pleistocene (Table S1). Applying the range-through assumption (i.e., a genus is present in the entire interval between its first and last known occurrence) yields similar results (Fig. 1): two for the Late Cretaceous, one for the Paleocene, four for the Eocene, two for the Oligocene, four for the late Oligocene—early Miocene, 11 for the Miocene, 13 for the Pliocene, and 14 for the Pleistocene. All modern majoid families are represented except for the Hymenosomatidae, which do not have a fossil record. Most majoid and all mithracid genera are found in the Caribbean region as opposed to in higher latitudes (Fig. 2). Thus far, eight genera are known from the Miocene—Pleistocene deposits from the Caribbean, whereas no more than three genera have been found in all other regions. An analysis of species diversity shows very similar numbers, with an apparent diversity hotspot in the Caribbean (Table 2).

Figure 1 Stratigraphic ranges of all families and genera of spider crabs (Majoidea) in the Western Atlantic based on occurrence data (Table S1).

Grey bars represent probable occurrences based on modern or bracketing fossil occurrences for that taxon. Chart arranged stratigraphically and by family. The Hymenosomatidae have no fossil record and the Priscinachidae are only known from Europe thus far. The ranges of families are derived from genera; genus names that were uncertain (aff., ?[genus], or “[genus]”) were not used. Timescale produced with TSCreator 6.4 (http://www.tscreator.org).

Figure 2 Genus-level diversity of all fossil spider crabs in the Western Atlantic from older to younger epochs (A–D) based on occurrence data (Table S1).

Genera of Mithracidae s.s. are indicated as gray parts of pie where present. Genus names that were uncertain (aff., ?[genus], or “[genus]”) were not included. Geographic regions were defined as follows: Atlantic coast North America (Canada to Georgia); Gulf of Mexico (incl. Florida); Caribbean (Cuba to Panama to Barbados); Atlantic coast South America (Guyana and further south). The youngest epoch was arbitrarily chosen for genera that could be either from one epoch or the following. No records are known from the Paleocene and Oligocene (range-through assumption not applied here).

Table 2 Species-level diversity of all fossil spider crabs in the Western Atlantic based on occurrence data (Table S1).

Species of Mithracidae s.s. are indicated between brackets. The youngest epoch was arbitrarily chosen for species that could be either from one epoch or the following.

	Pleistocene	Pliocene	Miocene	Oligocene	Eocene	Paleocene	Late Cretaceous	
Atlantic coast North America (Canada–Georgia)	3	4	1		3		1	
Gulf of Mexico Region (incl. Florida)	3 [1]	2	1		1		1	
Caribbean Region (Cuba–Panama–Barbados)	9 [7]	8 [5]	9 [7]		1			
Atlantic coast South America (Guyana and further south)			3					

Discussion

All modern majoid families are represented in Fig. 1 except for the Hymenosomatidae that do not have a fossil record. This is likely related to their small size and weakly calcified exoskeleton (e.g., Ng & Jeng, 1999; Guinot, 2011; Tavares & Santana, 2015; note that Guinot argued that the family does not belong to the Majoidea). Conversely, the Epialtidae and Mithracidae are well-represented, being markedly larger and better calcified, comparatively. Although many gaps exist in the fossil record of Western Atlantic majoids, it becomes clear that most families were present in the Western Atlantic in the Late Cretaceous to Paleogene. Given the oldest occurrences of majoids (Breton, 2009; Klompmaker, 2013), it may be speculated that spider crabs originated in Europe after which they crossed the proto-Atlantic to inhabit the Americas. Conversely, Mithracidae s.s. might have originated in the Western Atlantic because the globally oldest confirmed occurrences thus far are early Miocene in age and from that region (Tables 1 and S1).

Although the pattern that most fossil majoid genera in the Western Atlantic are found in the Caribbean region (Fig. 2) is consistent with the modern latitudinal diversity gradient for decapods, including Brachyura (e.g., Abele, 1982; Steele, 1988), much more research has been done in the (sub)tropical Western Atlantic region and exposures may be more numerous. However, fossil decapods from the eastern coast of the USA have received considerable attention (e.g., Rathbun, 1935; Roberts, 1962; Blow & Manning, 1996; Blow, 2003; Feldmann et al., 2013; Franţescu, 2013). On the other hand, less research has been done on fossil decapods from Brazil and other South American countries south of the Caribbean region thus far (e.g., Aguirre-Urreta, 1990; Casadío et al., 2005; Martins-Neto & Dias Júnior, 2007; Távora, Paixão & Da Silva, 2010). More fossil decapods—including spider crabs—are expected to be present in those regions given the common presence of majoids there today (e.g., Melo, 1996; Melo, 1998; Coelho & Torres, 1990; Bertini, Fransozo & De Melo, 2004; Mantelatto et al., 2004; Alves et al., 2012; Giraldes, Coelho Filho & Smyth, 2015).

The spider crabs Mithrax arawakum sp. nov. and Nemausa windsorae sp. nov., erected below, add to the number of decapod species known from the lower Miocene limestones at the Duncans Quarry in Jamaica. Portell & Collins (2004) reported on 16 brachyuran decapod species from these limestones, a unique fauna from the Miocene of the Caribbean because 9/14 genera were previously unknown from that region. Moreover, the Duncans Quarry is also of great importance for the knowledge of majoids in the Western Atlantic, yielding four genera and five species thus far, all mithracids. This is half of the majoid genera of the Miocene in the Caribbean region. As is the case for another diverse fossil brachyuran assemblage in the Caribbean, from the Pleistocene of Barbados (Collins & Morris, 1976), the brachyuran fauna from the Duncans Quarry is also associated with corals. This pattern of highly diverse decapod assemblages associated with corals is also observed elsewhere. Cenozoic, coral-associated decapod faunas from Europe are also speciose (e.g., Müller, 1984; Jakobsen & Collins, 1997; Beschin et al., 2007; Gatt & De Angeli, 2010; Beschin, Busulini & Tessier, 2015) as are such decapod faunas from the Mesozoic (e.g., Collins, Fraaye & Jagt, 1995; Fraaije, 2003; Krobicki & Zatoń, 2008; Klompmaker et al., 2013; Klompmaker, Ortiz & Wells, 2013; Robins, Feldmann & Schweitzer, 2013). Moreover, a significant positive correlation exists between global reef abundance and decapod diversity throughout the Mesozoic (Klompmaker et al., 2013). The diverse decapod assemblage from the Duncans Quarry underlines the importance of reefs for the abundance and diversity of decapods in deep time.

Systematic Paleontology

Order Decapoda Latreille, 1802–1803	
Infraorder Brachyura Linnaeus, 1758	
Section Eubrachyura De Saint Laurent, 1980	
Superfamily Majoidea Samouelle, 1819	
Family Mithracidae MacLeay, 1838	

Maguimithrax gen. nov.

Etymology—Combination of part of the family name of Tobey Maguire, the actor in three Spider-Man movies (2002, 2004, 2007), and Mithrax. Gender masculine.

Type species—Maia spinosissimusLamarck, 1818, by present designation, gender masculine, extant.

Species included—Maguimithrax spinosissimus (Lamarck, 1818).

Material—UF 12474 (1♀), 11447 (1♂), 11457 (1♀), 31157 (1♂, 1♀), 11388 (1♂, 1♀), all FLMNH IZ collection.

Diagnosis—Carapace slightly longer than wide to about equally wide as long in large specimens (l/w ratio = ∼ 1.09–0.97) (Fig. 3), maximum reported width without spines 167 mm, rounded to diamond-shaped, without angled transition from antero- to posterolateral margin, covered with spines laterally and tubercles more axially. Upper orbital margin with four to five spines including strong outer orbital spine and axialmost spine; four suborbital spines including two spines on antennal article, axialmost one strongest. Lateral margin bears six spines, anteriormost ones with accessory spines at anterior bases, fifth and sixth spines weaker. Gastric, cardiac, and uro-metagastric regions surrounded by pronounced grooves; other regions less delineated. Chelipeds and other appendages spinose dorsally, less so to smooth ventrally; cheliped propodus with tubercles or spines on upper margins and two to four tubercles on inner side.

Figure 3 Carapace length/width ratio vs. log2 carapace width (mm) for extant Maguimithrax spinosissimus (Lamarck, 1818).

Maximum length was determined without the rostral spines and width was measured without the anterolateral spines. Trend line is logarithmic (y = − 0.08ln(x) + 1.3624). Data in Table S2.

Remarks—Verrill (1908), Rathbun (1925), and Wagner (1990) all noted that young specimens of D. spinosissimus are close to Nemausa acuticornis and N. cornuta. There are indeed many similarities between Nemausa and D. spinosissimus including the spinose character of the carapace and appendages, a comparable third maxilliped (see Windsor & Felder, 2014: Fig. 4), a longer than wide carapace in younger individuals (Fig. 4), and a similar groove and region structure of the carapace. Not surprisingly, D. spinosissimus has been placed in Nemausa (Coelho & Torres, 1990). Several differences exist compared to Nemausa as currently defined. The carapace is more rounded to diamond-shaped compared to the pyriform carapaces of Nemausa so that the point of maximum width is reached more anteriorly; D. spinosissimus bears six lateral spines, whereas Nemausa bears five such spines; and the spine at the lateral angle is very strong in Nemausa compared to other lateral spines, but it is less prominent than others in D. spinosissimus. Molecular phylogenetics support the assertion that D. spinosissimus does not fit within Nemausa (Windsor & Felder, 2014).

Figure 4 Dorsal and ventral views of modern male specimens of Maguimithrax spinosissimus that differ in size.

(A, B) UF 11447, Florida, USA; (C, D) UF 11388, Florida, USA (largest specimen). Note the difference in length/width ratios of the carapace. Scale bar width = 30 mm.

The species has been assigned to Mithrax as well (e.g., Provenzano & Brownell, 1977; Wagner, 1990). However, Mithrax, as currently defined, is markedly different in that (1) the third maxilliped endopod merus distomesial margin has a deep, angular excavation at the articulation with the palp in D. spinosissimus, whereas this merus exhibits no pronounced concavity in Mithrax (cf. Windsor & Felder, 2014); (2) the ornamentation of the carapace is more varied in Mithrax, consisting of more granules; (3) the propodus bears tubercles and spines in the examined specimens of D. spinosissimus, but it is smooth in Mithrax; and (4) molecular phylogenetics separates D. spinosissimus from Mithrax (Windsor & Felder, 2014).

Most recently, the latter authors assigned the species to Damithrax. However, it should be noted that D. spinosissimus is much more spinose on the dorsal carapace than other species of Damithrax (e.g., Desbonne & Schramm, 1867: pl. 8; Rathbun, 1925: pl. 135), including the type species. Moreover, the propodus is not smooth in D. spinosissimus unlike in other species of the genus, and specimens across a considerable size range (<75 mm carapace width) are slightly longer than wide or about equally wide as long, unlike the diagnosis of the genus. Not surprisingly, the species plots as a sister taxon to all other modern Damithrax spp. (Windsor & Felder, 2014: Fig. 2); the latter authors also indicated that this taxon “is somewhat the outlier” (p. 155). Finally, all three of the discussed genera possess a lateral angle of the carapace, whereas this area is much more rounded in D. spinosissimus. Thus, we erect a new genus to accommodate D. spinossisimus: Maguimithrax gen. nov.

Detailed descriptions of the species and ontogenetic variations were detailed by Rathbun (1925), Williams (1984), and Wagner (1990) that need no repeat here. Sexual dimorphism is evident in that larger males (>∼60 mm carapace width based on the studied material) exhibit a pronounced tooth on the occlusal surface of the dactylus, whereas females do not bear such a tooth.

Stratigraphic and geographic range—Extant only, North Carolina—Venezuela (Williams, 1984; Wagner, 1990).

Damithrax Windsor & Felder, 2014

Type species—Mithrax pleuracanthusStimpson, 1871, extant.

Species included—Damithrax hispidus (Herbst, 1782–1804) [=Maia spinicincta Lamarck, 1818; Mithrax laevimanus Desbonne in Desbonne & Schramm, 1867; Mithrax depressus Milne-Edwards, 1873–1880 (part); Mithrax caribbaeus Rathbun, 1920; Mithrax carribbaeus, Ng, Guinot & Davie, 2008 (incorrect spelling)]; Damithrax pleuracanthus (Stimpson, 1871); Damithrax tortugae (Rathbun, 1920); Damithrax unguis (Portell & Collins, 2004).

Emended diagnosis—Carapace wider than long [for large specimens, about equally long as wide for small specimens], overall shape pyriform; dorsal surface smooth to tuberculate, not obviously setose; [five] lateral spines or teeth, first two commonly with accessory spine, lateral angle with single spine; posterior margin tuberculate. Rostral horns blunt, sparsely setose, tips not converging, not reaching [far] beyond first movable article of antenna. Antenna fused basal article very broad, forming floor of orbit, bearing two or three blunt marginal spines or teeth, anteriormost the largest, decreasing posteriorly (third often very low, or not developed), anterior two visible in dorsal view. Orbit complete, dorsal margin weakly armed behind strong pre-ocular tooth, eyestalk protected above by single blunt dorsal tooth or tubercle separated by closed fissure from two or three blunt post-ocular teeth or tubercles. Third maxilliped endopod merus distomesial margin deeply, angularly excavated at articulation with palp. Cheliped greater than or equal to carapace length; merus dorsal surface spinous, spines not laminar; carpus varied from smooth to rough; propodus smooth; dactylus with enlarged proximal tooth when mature, opposed margins of fingers otherwise crenulate. Pereiopods two to five (ambulatory legs) decreasing in size anterior to posterior; articles finely setose; merus dorsal surface bearing large tubercles and spines, ventral surface with one to six tubercles or spinules; carpus dorsal surface spinous; propodus without spination; dactylus strong, approximately half length of propodus, dactylar lock well developed (adapted after Windsor & Felder, 2014, changes in brackets).

Remarks—The diagnosis of Windsor & Felder (2014) mentioned that the carapace is wider than long. While this generally applies to large specimens, small specimens can be about equally long as wide or even slightly longer than wide (Fig. 5).

Figure 5 Carapace length/width ratio vs. log2 carapace width (mm) for Damithrax unguis (Portell & Collins, 2004) from the lower Miocene of Jamaica vs. modern Damithrax hispidus (Herbst, 1782–1804) from Florida.

Maximum length was determined without the rostral spines and width was measured without the anterolateral spines. Trend lines are logarithmic (y = − 0.113ln(x) + 1.2088 for D. unguis; y = − 0.072ln(x) + 1.191 for D. hispidus). Data in Table S2.

Damithrax unguis (Portell & Collins, 2004)

Figs. 5 and 6

Figure 6 Growth series of dorsal carapaces of Damithrax unguis (Portell & Collins, 2004) from the lower Miocene coral-associated limestones of the Montpelier Formation in the Duncans Quarry, Jamaica.

(A) = is RTV silicone rubber cast of external mold. (B–K) = internal molds. (A) UF 255051; (B) UF 113677; (C) UF 106768 (paratype); (D) UF 255053; (E) UF 112795; (F) UF 112783; (G) UF 112784; (H) UF 106697 (holotype); (I) UF 103954; (J, K) frontal and left-lateral views of UF 113677. Scale bar below (B) applies to (A–H). Scale bar width = 10.0 mm.

2004 Mithrax unguis sp. nov.; Portell & Collins, 2004: p. 117, Fig. 1.6.

Locality—FLMNH-IP XJ015: Duncans Quarry 01, Trelawny Parish, Jamaica (18.4710, −77.5796 WGS 84).

Stratigraphic horizon—lower Miocene, Montpelier Formation (uppermost unit) (Mitchell, 2004; Portell & Collins, 2004).

Material—Holotype: UF 106697; Paratypes: UF 73089, 73165, 103955, 106768, 106772, 111483; Topotypes: UF 112783–112785, 112795, 112942, 112946, 113010, 113011, 113117, 113586, 113587, 113675, 113677, 255051–255054. All internal molds of carapaces, some RTV silicone rubber casts of external molds of carapaces.

Diagnosis—Pyriform carapace, l/w ratios vary from ∼0.90 for the largest specimens of ∼16 mm width, to ∼1.00 for small specimens. Short rostrum with two small spines downturned, slightly longer than axialmost inner orbital spine. Four usually single spines (second one may have accessory small spine anteriorly in some specimens) on anterolateral margin excluding outer orbital spine. Forwardly directed shallow orbit with spines on the upper orbital margin: four upper orbital spines including outer orbital spine with center two converging; suborbital margin with three spines, axialmost one strongest. Smaller orbital spines less pronounced in small specimens. Tubercular gastric and branchial regions.

Description—See Portell & Collins (2004: p. 117).

Measurements— Table S2.

Remarks—Portell & Collins (2004) erected Mithrax unguis based on early Miocene specimens from the Duncans Quarry, Trelawny Parish, Jamaica. The generic placement was reassessed here because of the revision of extant Mithracidae by Windsor & Felder (2014). Given the close similarity to Damithrax hispidus, as was also indicated by Portell & Collins (2004), and a reasonable fit with the current generic diagnosis of Damithrax, Mithrax unguis is transferred to Damithrax. The species cannot be retained in Mithrax because of the non-spinose character on the dorsal carapace not including the lateral margins. The species differs from D. hispidus, D. pleuracanthus, and D. tortugae in that the rostrum is sharp instead of blunt and that D. unguis seems to have sharper upper orbital spines. Moreover, the length/width ratios separate D. unguis from D. hispidus (Fig. 5).

Portell & Collins (2004) had a limited number of specimens available and showed measurements for three of them. With additional collecting, preparation, and identification, many new specimens became available allowing for the investigation of ontogenetic variation within the species. Width grows faster relative to the length resulting in a decline of length/width ratios (Figs. 5 and 6). Such allometric growth is especially important for genera of Mithracidae that are currently diagnosed, in part, based on carapace length/width ratios (Windsor & Felder, 2014). For D. unguis, one could postulate that width is greater than length for some, width is (sub)equal to length, and even length is greater than width for the smallest specimens. Therefore, providing a range of l/w ratios along with specimen sizes for diagnoses and descriptions seems even more useful.

Stratigraphic and geographic range—lower Miocene, Jamaica.

Damithrax cf. pleuracanthus (Stimpson, 1871)

Figs. 7–10

Figure 7 Damithrax cf. pleuracanthus from the late Pliocene–early Pleistocene of the MacAsphalt Shell Pit, Sarasota County, Florida, USA (UF 29057).

(A) Dorsal view; (B) Ventral view; (C) Frontal view; (D) Right-lateral view; (E) Left-lateral view. Scale bar width = 10.0 mm.

Figure 8 Dorsal views of modern specimens and a single fossil specimen of Damithrax spp., all from Florida, USA.

Upper row from left to right—modern D. hispidus: UF 12475, 11604, 1082, 1086; Middle row—modern D. pleuracanthus: UF 3673, 9588 (largest specimen of lot), 7874, 1052; lower row—fossil Damithrax cf. pleuracanthus: UF 29057. Scale bar width = 10.0 mm.

Figure 9 Posterior views of similar-sized, modern specimens and a single fossil specimen of Damithrax spp.

(A) D. hispidus: UF 1082; (B) D. pleuracanthus: UF 7874; (C) Damithrax cf. pleuracanthus: UF 29057. For specimen sizes see Fig. 8.

Figure 10 Frontal views of similar-sized, modern specimens and a single fossil specimen of Damithrax spp.

(A) D. hispidus: UF 1082; (B) D. pleuracanthus: UF 7874; (C) Damithrax cf. pleuracanthus: UF 29057. For specimen sizes see Fig. 8.

Locality—FLMNH-IP SO001: MacAsphalt Shell Pit, Sarasota County, Florida, USA (27.3666, −82.4520 WGS 84).

Stratigraphic horizon—late Pliocene–early Pleistocene, spoil (float).

Material—Single carapace (UF 29057), cuticle.

Diagnosis—See Williams (1984: p. 334, 335).

Description—Carapace pyriform, about as long as wide (l/w ratio = 1.01), maximum width at ∼61% of carapace length, weakly convex longitudinally and moderately so transversely. Rostrum with two forward projections, only bases preserved; with blunt triangular axial projection oriented downward and posteriorly, with rims. Orbits directed anterolaterally, about as wide as tall, deep, with seven spines around orbit: two spines on antennal segment of which the axialmost one is strongest, separated by a notch and then followed by weak spine more laterally; upper orbital margin with four spines including strong outer orbital spine and stronger axialmost spine; two weak spines in between. Circular antennal holes between axialmost suborbital spine and rostral spines. Anterolateral margin with four spines (excluding outer orbital spine), third spine weakest, last spine at transition from antero- to posterolateral margin, oriented laterally. Posterolateral margin more rounded than anterolateral margin, with single small spine just posterior to previous spine. Posterior margin with convex protrusion axially, exhibiting row of tubercles and granules continuing onto posterolateral margin. Frontal region including epigastric region with double row of tubercles. Hepatic regions small, at lower level compared to gastric region, with single anterolateral spine. Protogastric regions bulbous, with major tubercle laterally and less pronounced tubercle axially. Mesogastric region with tubercle on process; base swollen, divided into three regions, central region oval. Uro- and/or protogastric region small, wider than long. Cardiac region pentagonal to triangular, with concave margins, about equally long as wide, tubercular. Branchial regions confluent. Intestinal region not delineated, with two strong tubercles. Cervical groove moderately deep, with two slits axially, V-shaped overall but rounded axially, bends more laterally near anterolateral margin. Shallow groove extends from cervical groove near base hepatic region to below outer orbital spine. Grooves around cardiac and uro- and/or metagastric regions. Dorsal carapace surface of cuticle with very small pits, armed with tubercles all over, more granules posteriorly; row of five tubercles in center of gastric region. Ventrolateral sides below anterolateral margins contain small spines. Of hardened parts: most of ventral surface, abdomen, and appendages lacking.

Measurements—Excluding spines and rostrum: 13.9 mm long, 13.8 mm wide.

Remarks—The specimen is very well-preserved and is ascribed to Damithrax because of the close similarity to extant species, notably Damithrax hispidus, D. pleuracanthus, and D. tortugae. These modern species were synonymized by Wagner (1990), but Windsor & Felder (2009) resurrected them based on molecular evidence and supported by morphological characters of the appendages. Ornamentation on the dorsal carapace, as was used by Rathbun (1925), was rejected by Windsor & Felder (2009) because of ontogenetic variability (accessory spines and tubercles become more apparent with age), especially within D. pleuracanthus. Ontogenetic variability of tubercles on the dorsal carapace was also found for D. hispidus in that the largest specimen (75.4 mm carapace width) exhibits fewer tubercles compared to small specimens (<∼30 mm carapace width) (A Klompmaker, pers. obs., 2015). Windsor & Felder (2009) suggested that ornamentation on the merus and carpus of the cheliped can be used to distinguish between D. hispidus, D. pleuracanthus, and D. tortugae. The FLMNH IZ collection contained sufficient specimens of D. hispidus and D. pleuracanthus to verify identifications. Indeed, specimens of D. pleuracanthus contain more tubercles on the carpus, but ornamental differences were difficult to verify for the merus. While large specimens of D. hispidus (>∼35 mm carapace width) often contained two spines on the inner side of the merus, smaller specimens (<∼23 mm carapace width) often contained only a single tubercle, much like similar-sized specimens of D. pleuracanthus (Table S3). An additional character to distinguish the two species is ornamentation on the dorsal carapace: tubercles appear better developed on the branchial and gastric regions of D. pleuracanthus relative to D. hispidus (Figs. 8–10). These differences are confirmed for slightly larger specimens from Rathbun (1925: pls. 146.1, 150.1), whereas D. tortugae appears to have even coarser dorsal tubercles (Rathbun, 1925: pl. 147.2). Additionally, a row of small tubercles is present along the posterolateral margin in D. pleuracanthus, but is absent in D. hispidus for the examined size range (Fig. 9). Thus, we argue that ornamentation on the dorsal carapace can be used to distinguish among modern species for similar-sized specimens. The fossil specimen conforms best to D. pleuracanthus in terms of coarseness of the tubercles and the presence of a row of small tubercles along the posterolateral margin. Given the lack of chelipeds to confirm species placement and some minor differences that may represent intraspecific variability (e.g., less robust anterolateral spines in the fossil specimen), the ascription is with some query. Nevertheless, this is the first record of this species in the fossil record. The results in Windsor & Felder (2009) and herein suggest that the ascription of fossil specimens to D. hispidus (e.g., Collins & Morris, 1976; Morris, 1993; Collins, Donovan & Dixon, 1996; Varela & Rojas-Consuegra, 2011) may need to be revisited.

Stratigraphic and geographic range—late Pliocene–early Pleistocene to Recent, North Carolina—Venezuela—Bermuda (Williams, 1984, see also Tavares & Albuquerque, 1993).

Mithrax Milne-Edwards, 1873–1880

Type species—Cancer aculeatusHerbst, 1782–1804 (see Windsor & Felder, 2014), extant.

Species included—Mithrax aculeatus (Herbst, 1782–1804) [=Cancer spinosus (Herbst, 1782–1804); Cancer aculeatus Fabricius, 1793; Mithrax pilosus Rathbun, 1892; Mithrax verrucosus Milne-Edwards, 1832; Mithrax plumosus Rathbun, 1901; Mithrax trispinosus Kingsley, 1879]; Mithrax armatus De Saussure, 1853 [=Mithrax orcutti Rathbun, 1925]; Mithrax arawakum sp. nov.; Mithrax bellii Gerstaecker, 1857; Mithrax besnardi Melo, 1990; Mithrax braziliensis Rathbun, 1892; Mithrax caboverdianus Türkay, 1986; Mithrax clarionensis Garth, 1940; Mithrax hemphilli Rathbun, 1892; Mithrax leucomelas Desbonne in Desbonne & Schramm, 1867; Mithrax tuberculatus Stimpson, 1860.

Diagnosis—See Windsor & Felder (2014: p. 162, 163).

Mithrax arawakum sp. nov.

Fig. 11

Figure 11 Type specimens of Mithrax arawakum sp. nov. from the lower Miocene coral-associated limestones of the Montpelier Formation in the Duncans Quarry, Jamaica.

(A, D, E) Holotype, UF 112682, in dorsal, frontal, and left-lateral views, resp.; (B) Paratype, external mold, UF 112941; (C) Paratype, cast of external mold, UF 112941. Scale bar width = 10.0 mm.

Etymology—Named in honor of the Arawak natives, who settled the island of Xaymaca (Jamaica).

Type material—UF 112682 (holotype, internal mold), UF 112941 (paratype, external mold + RTV silicone rubber cast).

Type locality—FLMNH-IP XJ015: Duncans Quarry 01, Trelawny Parish, Jamaica (18.4710, −77.5796 WGS 84).

Type horizon—lower Miocene, Montpelier Formation (uppermost unit) (Mitchell, 2004; Portell & Collins, 2004).

Material—No material known other than type specimens.

Diagnosis—Carapace pyriform, slightly longer than wide (l/w ratio = 1.03 for holotype). Short rostrum with two small spines downturned. Orbits directed forward, with at least four distinct spines around orbit: one long spine at angle of suborbital margin near rostral horns, other such spines not preserved; a slender and long outer orbital spine; a small central upper orbital spine; and a large projection on upper margin near rostral horns. Anterolateral margin with four strong spines (excluding outer orbital spine), middle two with small spine at anterior base; last spine at transition from antero- to posterolateral margin, oriented laterally. Posterolateral margin more rounded than anterolateral margin, with single small spine just posterior to previous spine. Frontal region with two longitudinal rims connecting to rostral spines and tubercular epigastric regions. Cervical groove deep and wide, U-shaped. Branchiocardiac groove strongest around cardiac region, weaker more laterally. Dorsal carapace surface armed with tubercles, granules, and spines (especially on branchial regions), not very densely so.

Description—Carapace pyriform, slightly longer than wide (l/w ratio = 1.03 for holotype), maximum width at ∼65% of carapace length, weakly convex longitudinally and moderately so transversely. Short rostrum with two small spines downturned. Orbits directed forward, wider than tall, not very deep, at least four distinct spines around orbit: one long spine with a smaller spine axially at angle of suborbital margin near rostral horns, other such spines not preserved (may be broken); a slender and long outer orbital spine; a small central upper orbital spine; and a large projection on upper margin near rostral horns. Single small spine present below orbit. Anterolateral margin with four strong spines (excluding outer orbital spine), middle two with small spine at anterior base; last spine at transition from antero- to posterolateral margin, oriented laterally. Posterolateral margin with single small spine just posterior to previous spine. Posterior margin with convex protrusion axially, with row of granules adjacent to convexity. Frontal region with two longitudinal rims connecting to rostral spines and tubercular epigastric regions. Hepatic regions small, at lower level compared to gastric region, with single strong anterolateral spine. Protogastric regions bulbous, with major tubercle laterally and less pronounced one axially. Mesogastric region with tubercle on process; base swollen, divided into three regions. Uro- and/or protogastric region small, appears as a laterally elongated tubercle. Cardiac region pentagonal, about equally long as wide, tubercular. Branchial regions weakly divided; epi- and mesobranchial regions confluent, tubercular; metabranchial separated from others, with spines, tubercles, and granules. Intestinal region not delineated, with two strong tubercles. Cervical groove deep and wide, U-shaped, bends more laterally near anterolateral margin to continue on ventral carapace, where it bends forward. Short groove extends from cervical groove near base hepatic region to outer orbital spine. Branchiocardiac groove strongest around cardiac region, weaker more laterally, not expressed to very weak on ventral carapace. Dorsal carapace surface armed with tubercles, granules, and spines (especially on branchial regions), not very densely so; row of five tubercles in center of gastric region. Of hardened parts: most of ventral surface, abdomen, cuticle, and appendages lacking.

Measurements—Excluding spines and rostrum: 14.0 mm long, 13.6 mm wide (UF 112682); length not measurable, 13.0 mm wide (UF 112941).

Remarks—The species appears to fit best in Mithrax because (1) the carapace is about equally long as wide (l/w ratio = 1.03); (2) the dorsal ornamentation has tubercles, granules, and spines (although less obvious than in most Mithrax spp.); and (3) the orbit is weakly produced and has two spines on the upper margin excluding the outer orbital spine.

The new species differs from all other congenerics. The carapaces of M. aculeatus, M. armatus, M. bellii, M. besnardi, and M. hemphilli exhibit a dense cover of granules (Rathbun, 1925: pls. 138.3, 139, 140, 142, 144; Garth, 1946: pl. 66, 1958: pl. 40.2; Melo, 1990; A Klompmaker, pers. obs., 2015 FLMNH IZ collection for M. aculeatus), whereas granules are much less abundant in the new species. Additionally, M. besnardi has a higher number of spines on the upper orbital margin (four excluding outer orbital spine instead of two). For M. braziliensis, Rathbun (1892) mentioned that the regions of this species are weakly defined, unlike the present species. Moreover, the upper orbital margin bears two small spines, whereas the new species bears one small and one larger one excluding the outer orbital spine. Although the ornamentation on the dorsal carapace of M. caboverdianus seems comparable (tubercles and spines with some interspersed granules) to the new species, the similar-sized holotype in Türkay (1986) (15.3 mm long) appears somewhat longer than wide (l/w ratio = 1.09) relatively (1.03 for Mithrax arawakum sp. nov.), but more specimens are needed to confirm this potential difference. Distinct rostral spines are missing in M. caboverdianus, but are present in Mithrax arawakum sp. nov. Additionally, the cardiac region in M. caboverdianus appears wider. The upper orbital margin contains more spines in M. clarionensis and the spines on the lateral margin are less prominent for a similar-sized specimen (Garth, 1940: pl. 15). Mithrax leucomelas was never figured and the specimen was already lost when Desbonne & Schramm (1867) erected the species. The description suggests that this species is different from the new species because M. leucomelas is said not to be spinose, the anterolateral margins are only slightly toothed, and the lateral angle does not bear a spine, unlike the specimens herein. Lastly, the new species is less tubercular than M. tuberculatus for a similar-sized specimen (Rathbun, 1925: pl. 151.1). Moreover, the rostral horns of M. tuberculatus are blunt; they are sharp in the new species.

This taxon is of special importance because it constitutes the oldest confirmed record of fossil Mithrax. The early Miocene record of Mithrax sp. from Cuba (Varela, 2013) is based on a fixed finger, which may not be sufficient for a genus ascription in light of the recent revision (Windsor & Felder, 2014). The same applies to other appendage fragments attributed to Mithrax sp. as well as incomplete carapaces (see Table S1).

The holotype is an internal mold, whereas the paratype is an external mold. Since the size of the two specimens is similar, the ornamentation can be compared. The cast of the external mold shows ornamentation that is largely the same to that of the internal mold, but some granules appear larger (those near the posterior margin).

Stratigraphic and geographic range—lower Miocene, Jamaica.

Nemausa Milne-Edwards, 1873–1880

Type species—Pisa spinipesBell, 1836, subsequent designation, extant.

Species included—Nemausa acuticornis (Stimpson, 1871); Nemausa cornuta (De Saussure, 1857) [=Nemausa rostrata Milne-Edwards, 1873–1880]; Nemausa donovani (Portell & Collins, 2004); Nemausa windsorae sp. nov.; Nemausa sinensis (Rathbun, 1892); Nemausa spinipes (Bell, 1836) [=Mithrax mexicanus Glassell, 1936].

Diagnosis—See Windsor & Felder (2014: p. 163, 164), but note that the now included fossil species and N. sinensis all have a tubercular rather than spinous character on the dorsal surface.

Remarks—Mithrax donovani (Fig. 12) is moved to Nemausa because the carapace is longer than wide in Nemausa, whereas the carapace length is subequal to the width or wider than long in the diagnosis of Mithrax (see Windsor & Felder, 2014). The small size of the specimen (6.7 mm maximum width, 8.0 mm preserved length excluding rostrum) suggests that not all characters may have fully developed yet (anterolateral spines, dorsal ornamentation, length/width trajectory), so the ascription to this genus is preliminary until better preserved material is discovered.

Figure 12 The holotype of Nemausa donovani (Portell & Collins, 2004) from the lower Miocene coral-associated limestones of the Montpelier Formation in the Duncans Quarry, Jamaica (UF 103958).

(A) Dorsal view; (B) frontal view; (C) angled right-lateral view; (D) upper view of rostrum and orbit; (E) right-lateral view. Scale bar width = 5.0 mm for (A–C, E); 1.5 mm for (D).

As for other spider crabs studied herein, ontogenetic change in the length/width ratios is evident for Nemausa as well (Fig. 13). The relationship for the species with the most specimens available, N. acuticornis, is best explained by a logarithmic trend line, suggesting that length/width ratios change faster in smaller specimens.

Figure 13 Carapace length/width ratio vs. log2 carapace width (mm) for Nemausa spp.

Nemausa donovani was not included because the total length could not be determined. Maximum length was determined without the rostral spines and width was measured without the anterolateral spines. Trend line is logarithmic (y = − 0.095ln(x) + 1.4013). Data in Table S2.

Nemausa windsorae sp. nov.

Figs. 13 and 14

Figure 14 The holotype of Nemausa windsorae sp. nov. from the lower Miocene coral-associated limestones of the Montpelier Formation in the Duncans Quarry, Jamaica (UF 113651).

(A) Dorsal view (internal mold); (B) dorsal view (cast of external mold); (C) frontal view; (D) right-lateral view; (E) external mold; (F) upper margin left orbit; (G) cast showing bases of rostral horns and various orbital spines in more detail. Arrows in (G) indicate suborbital spines and broken outer orbital spine. Scale bar width = 20 mm for (A–E); 2.0 mm for (F); 10 mm for (G).

Etymology—Named after Amanda M. Windsor for her work on extant majoids, especially mithracids.

Type material—Holotype and sole specimen, UF 113651 (internal mold with some cuticle, external mold + RTV silicone rubber cast).

Type locality—FLMNH-IP XJ015: Duncans Quarry 01, Trelawny Parish, Jamaica (18.4710, −77.5796 WGS 84).

Type horizon—lower Miocene, Montpelier Formation (uppermost unit) (Mitchell, 2004; Portell & Collins, 2004).

Material—No material known other than type specimen.

Diagnosis—Length/width ratio pyriform carapace = 1.19; orbital margins with seven spines, one long spine at angle of suborbital margin near rostral horns and two additional, smaller spines on same margin; anterolateral margin of carapace with four strong spines, anteriormost two with small spine at anterior base; mesogastric region flattened, anterior part not defined.

Description—Carapace pyriform, length/width ratio = 1.19, maximum width at 59% of carapace length, moderately convex longitudinally and transversely. Rostrum incompletely preserved, but with bases of two diverging spines. Orbits anterolaterally directed, wider than tall, deepest in most lateral part, seven spines around orbit: one long spine at angle of suborbital margin near rostral horns and two additional, smaller spines on same margin, separated by notch that marks boundary between antennal segment and rest of suborbital structure; one strong outer orbital spine with elongated base; three supraorbital spines, one closest to rostrum strongest. Anterolateral margin with four strong spines (excluding outer orbital spine), anteriormost two with small spine at anterior base; last strong spine at transition from antero- to posterolateral margin, directed laterally. Posterolateral margin more rounded than anterolateral margin, with single spine just posterior to previous spine. Gastric and hepatic regions mostly undifferentiated; epigastric regions appear as tubercles; base of mesogastric region swollen, anterior part not defined; uro- and/or metagastric region small, wider than long, sandwiched between mesogastric and cardiac regions. Cardiac region hexagonal. Branchial and intestinal regions confluent. Cervical groove deepest axially; curves around base mesogastric region, then becomes shallower and bends transversely to intersect lateral margin between first and second anterolateral spines. Branchiocardiac groove only defines lateral parts of cardiac region, does not reach lateral margin. Dorsal carapace surface armed with larger and smaller tubercles; row of five pronounced tubercles in center of gastric region; other strong tubercles present on epigastric, branchial, and cardiac regions. Of hardened parts: ventral surface, abdomen, and appendages missing; rostral spines largely missing.

Measurements—Excluding spines and rostrum: 27.7 mm long, 23.3 mm wide, and 14 mm tall (as preserved).

Remarks—The anterolateral spines are about equally prominent on the cast and the internal mold. The bases of the rostral spines and many of the orbital spines are much better seen on the cast. This is not surprising given the delicate nature of spines, which have the tendency to break easily on the internal mold. Perhaps surprisingly, the small tubercles on the dorsal carapace are not as numerous on the cast, yet another example that ornamentation with and without the cuticle can differ (see Lörenthey & Beurlen, 1929; Klompmaker, Hyžný & Jakobsen, 2015). Here, the difference can at least in part be explained by the fact that still some cuticle is present near/in those tubercles in the external mold, leading to the absence or less obvious tubercles on the cast.

Nemausa acuticornis is consistently more differentiated in the gastric region (e.g., center mesogastric region better defined and outlined: Fig. 14; Rathbun, 1925: pl. 136.1; Felder et al., 2014: Fig. 7C). Moreover, as mentioned by Rathbun (1925: p. 391), Fig. 15 shows that the suborbital margin of N. acuticornis contains only one pronounced spine between the outer orbital spine and the spines on the antennal segment, whereas this specimen bears two distinct spines there. Finally, N. acuticornis is relatively wider for specimens of the same size (Fig. 13).

Figure 15 Growth series of dorsal carapaces of modern Nemausa acuticornis (Stimpson, 1871) from various localities of the Atlantic coast of Florida, USA.

Note that specimens become relatively wider with age. (A) FSBC I-9758; (B) FSBC I-050561; (C) FSBC I-050562 (note the ‘unicorn’ rostrum instead of a double-horned rostrum); (D) FSBC I-050562; (E) FSBC I-050562; (F) FSBC I-050561; (G) FSBC I-050562; (H) FSBC I-050562. Scale bar width = 30 mm.

Nemausa cornutus exhibits more spinose ornamentation on the carapace (Rathbun, 1925: pl. 137.3 and 137.4) even though the specimens are larger (larger specimens tend to have weaker ornamentation compared to younger specimens from the same species in the Mithracidae). Moreover, the specimens in Rathbun (1925: pl. 137.3 and 137.4) are narrower (Fig. 13), although more specimens are needed to statistically test this difference.

Nemausa sinensis has a lower l/w ratio (1.03 (Garth, 1958: pl. 41.1), 1.06 (Rathbun, 1892: pl. 38.2)) compared to N. windsorae sp. nov. (1.19) (Fig. 13). Furthermore, stronger tubercles are present on N. sinensis.

Very few specimens of N. spinipes are figured, with Rathbun (1925) showing the best image. Nemausa spinipes has a better defined mesogastric region (Rathbun, 1925: pl. 136.4) and all anterolateral spines are single and not associated with smaller spines as in the specimen under study. The same author also showed a very strong tubercle on the posterior part of the mesogastric region, not seen in the specimen under study; and two instead of one tubercle are present around the location where the mesogastric process would be.

Nemausa donovani is different in that the mesogastric region is outlined entirely and a distinct elevation is seen in the center of the posterior part of this region, both unlike in the new species. This is unlikely to be related to ontogeny because the mesogastric features appear stable throughout ontogeny in a congeneric species (Fig. 15). Although anterolateral spines become more prominent throughout ontogeny in Nemausa (N. acuticornis, Fig. 15), the difference between N. donovani and N. windsorae sp. nov. is much greater, supporting the hypothesis that these are two separate species. Furthermore, N. windsorae sp. nov. bears a denser ornamentation of tubercles, which may only in part be explained by ontogeny (Fig. 15) because even the smallest specimen of N. acuticornis bears distinct tubercles on the branchial regions, whereas these regions are nearly smooth in N. donovani, unlike for N. windsorae sp. nov.

Stratigraphic and geographic range—lower Miocene, Jamaica.

Supplemental Information

Table S1 Database fossil majoids from the Western Atlantic.

Click here for additional data file.

Table S2 Measurements modern and fossil mithracids from the Western Atlantic.

Click here for additional data file.

Table S3 Data on Damithrax hispidus from Florida.

Click here for additional data file.

The reviewers Amanda Windsor (Smithsonian Institution) and Douglas Alves (Universidade Federal de Sergipe) as well as academic editor Kenneth de Baets are thanked for very useful comments that improved this manuscript. Also, we are grateful to Chad Walter and Rafael Lemaitre (USNM) for allowing examination of collections under their care; B. Alex Kittle and Sean Roberts (both FLMNH) for assistance in the collections at the USNM and FLMNH; Laura Wiggins, Robert Lasley, and Joan Herrera (Florida Fish and Wildlife Institute) for loan of N. acuticornis specimens used in this study; Douglas Jones and George Hecht (both FLMNH), the late Reed Toomey, and Julie Thaler for collecting assistance at Duncans Quarry; Amanda Windsor provided additional information on her work; Marcos Tavares (Museu de Zoologia, Universidade de São Paulo) provided items of literature; Cristina Robins (FLMNH) for inspiration for a taxon name; and the FLMNH Division of Invertebrate Zoology for access to modern spider crab specimens. The project was initiated during the course Paleontology (Fall 2015), University of Florida, Department of Geological Sciences. This is University of Florida Contribution to Paleobiology 694.

Additional Information and Declarations

Competing Interests

Author Contributions

New Taxa Registration

The authors declare there are no competing interests.

Adiël A. Klompmaker conceived and designed the experiments, performed the experiments, analyzed the data, contributed reagents/materials/analysis tools, wrote the paper, prepared figures and/or tables, reviewed drafts of the paper.

Roger W. Portell conceived and designed the experiments, contributed reagents/materials/analysis tools, prepared figures and/or tables, reviewed drafts of the paper.

Aaron T. Klier, Vanessa Prueter and Alyssa L. Tucker performed the experiments, analyzed the data, reviewed drafts of the paper.

The following information was supplied regarding the registration of a newly described species:

Zoobank:

Maguimithrax gen. nov.: http://zoobank.org/NomenclaturalActs/BE0D3F8C-7FD8-43A8-A42A-DE797B31A58E

Mithrax arawakum sp. nov.: http://zoobank.org/NomenclaturalActs/971193E9-B5BA-44EA-A889-C11B01B6C785

Nemausa windsorae sp. nov.: http://zoobank.org/NomenclaturalActs/2A8E6FEB-E453-4FFE-948B-237AE523652F.

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
