# Peer review of "Spider crabs of the Western Atlantic with special reference to fossil and some modern Mithracidae"

_PeerJ, doi:10.7717/peerj.1301_

## Round 0.1 · original submission · Minor Revisions

This is an interesting manuscript which considerably augments our knowledge about the fossil record of Majidae, particularly in the Western Atlantic. There are, however, some points which need to be improved before publication (see also comments by reviewers):

Fossil Record figures: Figure 1 and 2 should be discussed more explicitly (see comments by reviewer 1)

Size relationships/allometry: You allude to the importance of allometric growth and length/width ratios in the abstract, but do not discuss this in detail in the introduction or discussion. These are only discussed in Remarks of various species and nicely illustrated in Figure 3 and 13. I concur with reviewer 1 that it would be nice to introduced and discuss size relationships/allometry more thoroughly in the main text too. An alternative would be too omit them entirely (see comments by reviewer 2), but as this is one of the main arguments for the taxonomic changes you make, I do not consider the latter advisable.

Photographs: The photographs of figures 8-10 are of inferior quality; these specimens/figures should be presented in the same way (without shadows, black background) as the other figures (see comment by reviewer 1)

In addition to these points and the ones raised by the reviewers, please also address the following:

Line 27: please remove D. from cf. D. pleuracanthus as this is irrelevant here

Line 42: please cite a reference for this large range in carapace length

Line 203: I would write “Thus, we erect a new genus to accommodate D. spinossisimus.”

Line 305: I think “Damithrax cf. pleuracanthus” should suffice.

·

Basic reporting

The introduction and discussion omit any mention of allometric growth and length/width ratios which are alluded to as very important in the Abstract. While allometric growth is discussed in the Remarks under several species descriptions and there are 2 figures dedicated to the topic, it should be addressed in the primary text as well.

Figures 1 and 2 are not sufficiently explained in the text or in the figure legends.
Figures 8 & 9 are of inferior quality to the preceding photographic figures

Other remarks are annotated within the PDF

Experimental design

No Comments

Validity of the findings

The reasoning behind the taxonomic changes being made here are well reasoned and supported by the data.
Again, the topic of allometric growth should be introduced and discussed more thoroughly.

Other comments are annotated within the attached PDF

Additional comments

Overall, this is a fine piece of work. However, there are many points that must be addressed more thoroughly to clarify the findings. The Remarks under each species description are very well reasoned, please apply the same level of detail to the introduction, results, and discussion.

·

Basic reporting

No comments.

Experimental design

No comments.

Validity of the findings

No comments.

Additional comments

The manuscript makes an important contribution to knowledge about the Majoidea. In particular on the fossil record along the western Atlantic.
I think the manuscript, in the current format, is extensive. In this way, the size relationships that were provided, greatly increase the size of the manuscript, but do not add new significant knowledge. I suggest that the graphics of crab size should be excluded.

---

## Round 0.2 · Minor Revisions

Your article is as good as accepted. There are just some minor issues which need to be resolved before I send the paper into production as these cannot be changed any more once I officially accept your paper:

Line 194: "Family Mithracidae Macleay, 1838" should be on the next line
Line 428: i would delete the "A" in "A. Milne-Edwards" as there can not be any confusion as their work was published in different years.
Line 544: i would delete the "A" in "A. Milne-Edwards" as there can not be any confusion as the work of H and A Milne-Edwards was published in different years.
Please check figures 7, 12 and 14 as there are some issues with contrast/resolution in the reviewing pdf. The original figures seem to be in order. I will also notify the PeerJ staff to pay attention to these issues.

---

## Round 0.3 · accepted · Accept

Thank you for making these final changes.